# Prognostic Markers of Microinvasive Breast Carcinoma: A Systematic Review and Meta-Analysis

**DOI:** 10.3390/cancers15113007

**Published:** 2023-05-31

**Authors:** Andrea Ambrosini-Spaltro, Francesco Di Donato, Luca Saragoni, Gábor Cserni, Emad Rakha, Maria Pia Foschini

**Affiliations:** 1Pathology Unit, Morgani-Pierantoni Hospital, AUSL Romagna, 47121 Forlì, Italy; andrea.ambrosinispaltro@auslromagna.it; 2Pathology Unit, Santa Maria delle Croci Hospital, AUSL Romagna, 48121 Ravenna, Italy; francesco.didonato6@studio.unibo.it (F.D.D.); luca.saragoni@auslromagna.it (L.S.); 3School of Anatomic Pathology, Department of Biomedical and Neuromotor Sciences, University of Bologna, 40139 Bologna, Italy; 4Department of Pathology, Bács-Kiskun County Teaching Hospital, 6000 Kecskemét, Hungary; csernig@kmk.hu; 5Department of Pathology, University of Szeged, 6725 Szeged, Hungary; 6Histopathology Department, School of Medicine, University of Nottingham, Nottingham NG7 2RD, UK; emad.rakha@nottingham.ac.uk; 7Unit of Anatomic Pathology, Department of Biomedical and Neuromotor Sciences, University of Bologna, Bellaria Hospital, 40139 Bologna, Italy

**Keywords:** microinvasive, breast, carcinoma, prognostic factor, systematic review, meta-analysis, lymph node metastasis

## Abstract

**Simple Summary:**

Microinvasive breast carcinoma is an infiltrating carcinoma that measures ≤1 mm. The prognostic factors associated with this disease have not been extensively investigated. In this study, we aimed to perform a systematic review and meta-analysis to examine the prognostic factors of microinvasive breast carcinoma. From 618 screened records, 5 were selected. The meta-analyses found a significant association between lymph node status and prognosis. No significant prognostic impacts were found for the following factors: estrogen receptor, progesterone receptor, HER2 status, multifocality or grade of microinvasion, or patient’s age. Therefore, the data obtained demonstrate that lymph node status is a main prognostic factor of microinvasive breast carcinoma.

**Abstract:**

(1) Background: The prognostic factors of microinvasive (≤1 mm) breast carcinoma are not completely clear. The aim of this study was to perform a systematic review and meta-analysis to clarify these factors. (2) Methods: The Preferred Reporting Items for Systematic Reviews and Meta-Analyses (PRISMA) methodology was followed. Two databases were interrogated, PubMed and Embase, and papers in English were included to address this question. The selected studies were those that reported on female patients affected by microinvasive carcinoma, and on prognostic factors with a hazard ratio (HR) for disease-free survival (DFS) and overall survival (OS). (3) Results: In total, 618 records were identified. After removing duplicates (166), identification, and screening (336 by title and abstract alone, 116 by full text and eventual supplementary material), 5 papers were selected. Seven different meta-analyses were conducted in this study, all referring to DFS, analyzing the following prognostic factors: estrogen receptor, progesterone receptor, HER2 status, multifocality and grade of microinvasion, patient’s age, and lymph node status. Only lymph node status was associated with prognosis and DFS (total number of cases: 1528; Z = 1.94; *p* = 0.05). The other factors examined did not significantly affect prognosis (*p* > 0.05). (4) Conclusions: Positive lymph node status significantly worsens prognosis in patients with microinvasive breast carcinoma.

## 1. Introduction

Microinvasive breast carcinoma is defined as infiltrating carcinoma not more than 1 mm in size [1]. It is typically found adjacent to ductal carcinoma in situ (DCIS), especially high-grade comedocarcinoma (Figure 1), but can also be associated with other types of ductal and lobular carcinoma in situ, including Paget’s disease of the nipple [2].

The size criteria for this entity are strict, and although the rules of rounding apply to the T categories of the TNM (tumor, node, and metastasis) classification, microinvasive carcinoma is an exception to this, and any cancer larger than 1 mm does not belong to this staging category. Currently, microinvasive carcinoma (pT1mi) can be diagnosed in excision specimens even in the absence of an identified in situ carcinoma component, if the size criterion is appropriate [3]. The incidence of microinvasive breast carcinoma varies from 0.68–2.4% [4] to 5% [5]. Conflicting results have been reported in terms of behavior. Some studies initially described a more favorable prognosis, similar to that of DCIS [6]. A meta-analysis of 2959 patents confirmed this view, stating that microinvasive carcinoma is not linked with higher rates of clinically significant metastasis to axillary lymph nodes, and its survival rates are very similar to those of DCIS [7]. However, the largest study conducted to date, a SEER analysis on 161,394 cases of DCIS and 13,489 cases of microinvasive carcinoma, highlighted that breast cancer-specific mortality rate was 3.8% for pure DCIS and 6.9% for microinvasive carcinoma, with the difference being significantly different [8]. The second largest series in the literature (11,285 pure DCIS and 521 microinvasive carcinoma) confirmed this more aggressive behavior: the presence of microinvasion was associated with significantly poorer breast cancer-specific mortality compared with that of patients with pure DCIS [5]. A recent meta-analysis showed that disease-free survival and locoregional recurrence-free survival were significantly shorter in microinvasive carcinoma than in pure DCIS; both overall survival and distant metastasis-free survival tended to be shorter even if without statistical significance [9]. For these reasons, microinvasive carcinoma is now considered more similar to small breast carcinoma (0.2–1.0 cm) than to pure DCIS [8], although it is less frequently associated with sentinel lymph node metastasis (from 2% [7] to 3.2% [10]). Nevertheless, the overall prognosis of microinvasive breast carcinoma is generally good. Studies on SEER registries showed that the 20-year cancer-specific mortality rates were 9.65% (4.00% in DCIS) [11] and 6.9% [8]. In a study on 1299 cases of microinvasive carcinoma, the 5-year locoregional-free survival, distant metastasis-free survival, and overall survival (OS) were 98.6%, 97.1%, and 99.4%, respectively [12]. The specific determinants of the biological behavior of microinvasive carcinoma have not been extensively investigated in the literature, and many studies have focused on comparing microinvasive carcinoma and pure DCIS. In breast carcinoma, prognostic factors are extremely useful for defining therapeutic options and establishing whether patients should undergo adjuvant treatment [13]. In traditional invasive carcinoma, histological grade, lymph node status, estrogen receptor (ER), progesterone receptor (PR), and proliferation by Ki67 and HER2 are well-defined prognostic and predictive factors [14]. In DCIS, prognostic factors include grade [15] and DCIS size [16]. The performance of these factors has not been extensively analyzed in microinvasive breast carcinoma, and further investigations on its prognostic factors could be relevant for clinical purposes. For these reasons, we aimed to perform a systematic review with meta-analysis to answer the following question: what are the prognostic factors of microinvasive breast carcinoma?

## 2. Materials and Methods

### 2.1. Guidelines and PICO

The present study followed the guidelines of the Preferred Reporting Items for Systematic Reviews and Meta-Analyses (PRISMA) 2020 statement [17]. The PRISMA checklist with the requested information is available in the Appendix A (Appendix A, PRISMA checklist). We described the PICO elements (population, intervention/index, comparison, and outcome) as follows:Participants: female patients with microinvasive breast carcinoma;Intervention/Index: prognostic factor examined;Comparison: not applicable;Outcome: disease-free survival (DFS), progression-free survival (PFS), or overall survival (OS).

### 2.2. Protocol Registration

Before starting the search, the present protocol was recorded on Prospero, a known portal for systematic reviews and meta-analyses. Registration number: CRD42022360089, available from: https://www.crd.york.ac.uk/prospero/display_record.php?ID=CRD42022360089 (accessed on 22 April 2023).

### 2.3. Search Strategy

On 5 October 2022, studies were searched using two databases, PubMed and Embase. The search strategy was based on the following question: What are the prognostic factors of microinvasive breast carcinoma? To answer this question, the two main search criteria were combined.

Microinvasive breast carcinoma;Prognosis.

We used a combination of keywords and a controlled vocabulary. The controlled vocabulary is composed of MeSH terms in PubMed and Emtree terms in Embase.

The following text was used for the two databases:3.PubMed((Microinvas* [tw]) AND (breast [tw] OR mammar* [tw]) AND (carcinoma [tw] OR cancer [tw]) OR (Microinvas* [tw] AND Breast Neoplasm [MeSH Terms]) OR (Microinvas* [tw] AND (breast [tw] OR mammar* [tw]) AND Carcinoma [MeSH Terms])) AND (“prognostic factor*” [tw] OR outcome [tw] OR survival [tw] OR Prognosis [MeSH Terms])

4.Embase(microinvas* AND (breast OR mammar*) AND (carcinoma OR cancer) OR (microinvas* AND “breast carcinoma”/exp)) AND (“prognostic factor*” OR outcome OR survival OR “prognosis”/exp)

For both PubMed and Embase searches, a weekly email alert was set to detect new entries on a regular basis. All new entries were recorded until 31 December 2022.

### 2.4. Selection of Articles

All citations obtained from the two different databases (PubMed and Embase) and their subsequent updates were imported into the online portal Rayyan, https://www.rayyan.ai (accessed on 11 March 2023).

Duplicates were suggested by the Rayyan portal and were all controlled by one of the authors (AAS). Two authors (AAS and FDD) independently selected the articles (some by abstract and title alone, some by entire full text, and eventually by supplementary material). In articles with disagreement, a consensus conclusion was reached for each case.

### 2.5. Eligibility Criteria

Articles based on the following inclusion criteria were selected:5.Female patients with invasive breast carcinoma not more than 1 mm in size.6.Documented prognostic impact of at least one of the following:a.Microinvasive carcinoma nuclear grade;b.DCIS grade;c.ER positivity, HER2 amplification in invasive carcinoma (via immunohistochemistry or in situ hybridization);d.DCIS extent;e.Unifocal vs. multifocal (more than 1) foci of microinvasion;f.Other prognostic factors were also recorded.7.The prognostic impact should have been documented by hazard ratio (HR) and corresponding 95% confidence intervals for DFS, PFS, or OS. HR values of prognostic factors had to be clearly expressed in the main text, tables, figures, or supplementary tables/material.8.English language.

Exclusion criteria; articles were not included when:9.Follow-up (FU) was not available.10.FU did not specifically address microinvasive carcinoma (e.g., considered together with DCIS).

### 2.6. Risk of Bias Assessment

The Quality In Prognosis Studies (QUIPS) tool was utilized to assess the risk of bias (RoB) [18]. The QUIPS tool is specifically designed for prognostic studies and is composed of 6 domains: (1) study participation, (2) study attrition, (3) prognostic factor measurement, (4) outcome measurement, (5) study confounding, and (6) statistical analysis and reporting. Each domain is classified into three groups: low, moderate, and high risk. Two authors (AAS and FDD) independently evaluated the selected articles. In articles with disagreement, a consensus conclusion was reached for each case.

### 2.7. Statistical Analysis

Hazard ratios (HRs) and 95% confidence intervals (CIs) were obtained from each article and for each prognostic factor. HRs were preferentially obtained from the multivariate analysis; if not available, HRs from the univariate analysis were considered. Meta-analyses and forest plots were performed using Review Manager (RevMan) software version 5.4 [19]. The RevMan software required HR values as natural logarithms, which were converted using the calculator tool provided by the software itself. Meta-analyses were conducted using the random effects model, which may better quantify the heterogeneity that is usually present and high in prognostic studies [20]. The heterogeneity of the results was assessed using *I*^2^ statistic output [21], which was directly calculated using the RevMan software. HRs from DFS were examined separately from HRs from OS. Statistical significance (*p*) was set at 0.05, using a 2-tailed hypothesis.

### 2.8. Quality of Evidence

To assess the overall quality of evidence, we used the Grading of Recommendations, Assessment, Development and Evaluation (GRADE) approach, specifically adapted for prognostic factors [22]. This approach is based on 5 main factors: (1) risk of bias, (2) inconsistency (or heterogeneity), (3) indirectness, (4) imprecision, and (5) publication bias. Each of the 5 main factors may determine to downgrade the overall quality by 1–2 levels.

## 3. Results

### 3.1. Article Selection

In total, 618 results were obtained. The PubMed search retrieved 250 records, with 4 new entries by email alert, resulting in 254 PubMed citations. The Embase search retrieved 361 records, with 3 new entries by email alert, resulting in 364 Embase citations. After removing 166 duplicates, 452 citations were obtained. Of these, 331 were excluded based only on title and abstract. In total, 121 citations were searched for full-text and 116 full-text articles were examined. Six articles were obtained from the same SEER database [11,23,24,25,26,27], and for this reason, only one of them could be selected; they examined only overall survival and cancer-specific survival as outcome measures, while all the other selected studies examined DFS. Moreover, studies from the SEER database may contain multi-institutional data with variations in the definition of microinvasive carcinoma, as evidenced by Shiino et al., who excluded them in a similar meta-analysis [9]. Therefore, all six articles obtained from the SEER database were excluded. The study conducted by Niu et al. was also excluded because it examined ER and PR together [28]. Ki67 was too variable with many cutoff values, so it was not considered in our meta-analyses. Finally, five articles were selected. Detailed information on the article selection is summarized in the PRISMA flow chart (Figure 2).

### 3.2. Risk of Bias Assessment

For risk of bias assessment, the QUIPS tool identified low-risk and moderate-risk categories. Detailed results are summarized in Table 1.

Fang [29] and Li [12] studied cases from the same city (Shanghai) in China, but they were retrieved from two different medical centers: Fang from Shanghai Jiaotong University School of Medicine, while Li from Fudan University Shanghai Cancer Center; therefore, the studies were considered separately in the present meta-analyses.

Rakovitch [32] collectively reported prognostic data for both microinvasive and DCIS; however, they also examined the impact of multifocality of invasion, which clearly only referred to microinvasive carcinomas. Therefore, from Rakovitch’s study, we were able to include only the impact of multifocality of microinvasion in our meta-analysis.

### 3.3. Population Examined

The selected articles examined a number of cases varying from 72 to 1286. The clinicopathological features of the five selected articles are summarized in Table 2.

### 3.4. Meta-Analysis Results

Seven different meta-analyses were performed. They addressed the following prognostic factors: ER, PR, HER2, presence of multiple foci of microinvasion (1 vs. ≥2), microinvasive grade (1/2 vs. 3), patient’s age (<50 vs. ≥50 years), and lymph node status. Because of different cut-off values, not all available articles could be included in the same meta-analysis; therefore, we chose the cut-off values more frequently used by the selected articles. The other prognostic factors could not be compared because of the completely different cut-off values among different studies: Ki67, margin status, DCIS size, and therapies.

Only lymph node status had a marginally significant effect on DFS (total number of cases: 1528; Z = 1.94; *p* = 0.05): positive lymph node status significantly worsened the prognosis (HR > 1) by reducing DFS. The other meta-analyses did not provide significant results: ER (*p* = 0.69), PR (*p* = 0.52), HER2 (*p* = 0.71), multiple foci of invasion (*p* = 0.24), grade (*p* = 0.44), and age (*p* = 0.38). The corresponding forest plots are shown in Figure 3. The main results of the meta-analyses are summarized in Table 3.

### 3.5. Quality of Evidence

The GRADE approach highlighted that the evidence quality of our meta-analyses was *moderate* for ER, PR, multiple foci, grade, and lymph node status, whereas it was *low* for HER2 and age (see Table 3). All studies included in our meta-analysis were observational and started with high certainty ratings using the GRADE approach for prognostic factors [22]. In all meta-analyses, the quality of evidence was downgraded by one level because of imprecision, that is, small size (limited numbers of articles with relatively wide confidence intervals; see Figure 3). In HER2 and age meta-analyses, the quality of evidence was downgraded by one more level for inconsistency (or heterogeneity), which was moderate to high (*I*^2^ > 50%); the meta-analyses of other prognostic factors showed low inconsistency (or heterogeneity) (*I*^2^ < 50%), so it was not necessary to downgrade them. In all meta-analyses, no high-risk element was found for risk of bias (see Section 3.2 and Table 1), indirectness of evidence, or publication bias.

## 4. Discussion

We believe that the present systematic review and meta-analysis provides the best possible evidence on prognostic factors for microinvasive breast carcinomas and suggests that lymph node status is the main prognostic parameter that can determine the outcome of the disease: patients with a positive lymph node status have a worse prognosis, with reduction in DFS. Conversely, the other factors examined (ER, PR, HER2, multifocality, grade, and age) did not significantly affect DFS. The GRADE approach for prognostic factors [22] highlighted that the quality of evidence in our meta-analyses was moderate for ER, PR, multiplicity of foci, grade, and lymph node status, whereas it was low for HER2 and age. This is mainly due to the high level of heterogeneity in the meta-analyses of HER2 and age.

Microinvasive breast carcinoma is a relatively rare disease, with an incidence varying from 0.68–2.4% [4] to 5% [5]. However, considering the wide diffusion of screening programs for breast carcinoma, the diagnosis of microinvasive breast carcinoma is not uncommon. Microinvasive carcinoma is typically associated with DCIS. In a study of 11,285 DCIS cases, microinvasion was identified in 4.6% (512) [5]. Conflicting results have been reported regarding prognosis. A study on 136 cases [6] and a meta-analysis on 2959 cases [7] reported a more favorable prognosis of microinvasive carcinoma, similar to that of DCIS. However, the two largest studies in the literature described a more aggressive behavior of microinvasive carcinoma in comparison with DCIS [5,8]. A recent meta-analysis confirmed that microinvasive breast carcinoma has a worse prognosis than pure DCIS [9]. For these reasons, it is more reasonable to admit that, unlike DCIS, microinvasive disease may represent an early-stage invasive carcinoma with metastatic potential. However, the specific prognostic factors of microinvasive breast carcinoma have not been extensively or uniformly examined in the literature. Many studies have analyzed the prognostic factors of microinvasive carcinoma (even with clearly expressed HRs), but these were collectively calculated for both CDIS and microinvasive carcinoma [6,8,33,34,35,36,37] and not specifically for microinvasive carcinoma. This represents a substantial knowledge gap, because defining the prognostic factors of a specific tumor is crucial for establishing a proper therapeutic strategy. 

In the present meta-analysis, several prognostic factors were analyzed, but only lymph node status affected prognosis (total number of cases: 1528; *p* = 0.05). Lymph node status is a strong prognostic factor in traditional invasive carcinoma and it directly affects the stage of the disease. In fact, if distant metastases are excluded, nodal status is still considered the most important prognosticator of invasive breast cancer [38]. It is reasonable to assume that lymph node status significantly affects the prognosis of microinvasive carcinoma, too. The detection of lymph node metastasis may identify the few cases with an expected worse prognosis, which may have a higher risk of progression and may be considered for a different therapeutic strategy. The presence of lymph node metastasis in microinvasive disease may reflect either an aggressive microinvasive carcinoma that has the ability to metastasize or the existence of an undetectable, larger invasive disease in the breast. 

However, the rate of sentinel node metastases in microinvasive carcinoma was low, affecting only 2.9% in a U.S. national study [39]. This was confirmed via two important meta-analyses. The first meta-analysis on 968 patients found that microinvasive breast carcinoma was associated with sentinel lymph node metastasis in a limited number of cases (3.2% for macrometastasis and 4.0% for micrometastasis) [10]. A recent meta-analysis on 2959 patients found that the pooled estimated sentinel node positivity rate specific for macrometastases was low (2%), even suggesting that sentinel lymph node biopsy is not required for the management of microinvasive carcinoma [7]. Unfortunately, the two selected articles in our meta-analysis on lymph node status did not specifically define whether lymph nodes were positive in terms of micrometastasis vs. macrometastasis; therefore, we were not able to better define this parameter.

The studies included in our meta-analysis highlighted several prognostic factors in their original reports. However, when considered together in the meta-analyses, only lymph node status was considered to retain its significance. Li et al. showed that patient’s age (>40 years) and close margins (≤2 mm) were independent prognostic factors in multivariate analysis, and lymph node status was significant in univariate analysis [12]. Pu et al. identified large DCIS size, lymph node involvement, and no radiotherapy as factors for worse DFS in microinvasive disease [31]. Si et al. analyzed many prognostic factors of microinvasive breast carcinoma, but they did not express HRs; therefore, their results were not included in the present meta-analyses. However, they also concluded that lymph node status was the only independent predictor of worse DFS [40].

Several reports suggest that, as a group, microinvasive carcinomas exhibit a peculiar biological profile, which differs from that of traditional infiltrating carcinoma. Mastropasqua et al. found that microinvasive carcinoma usually shows a different profile than small infiltrating carcinoma: microinvasive carcinoma is hormone receptor-positive in only 38.5% of cases, whereas it is more frequently HER2-positive (47%) [41]. In a multicenter study, Costarelli et al. reached the same conclusion, demonstrating that microinvasive carcinoma is more frequently ER-negative (35.8%), PR-negative (49.8%), and HER2-positive (38%) than usual infiltrating carcinoma [42]. Zhang et al. confirmed that microinvasive carcinoma frequently showed HER2 positivity (41.3%) with high-grade pathologic features [43]. These differences may reflect that HER2-positive invasive carcinomas more often derive from more extensive high-grade DCIS recognized on mammography due to their specific type of calcification [44]; therefore, the earliest step of their invasive life, i.e., microinvasion, is more often caught during the static histological examination of surgical specimens. Indeed, microinvasion is associated with high nuclear grade, comedonecrosis, and large extent, all features of worse prognosis [5]. Unfortunately, due to the very small size and rarity of this entity, it is very difficult to analyze its prognosticators. The more aggressive biology may be balanced by the small volume of the disease.

Although it is difficult to provide evidence-based data on the best treatment approach of microinvasive breast carcinoma due to the presence of many confounders that may impact on the outcome of patients such as the features of associated DCIS, the volume of invasive disease, which can be variable despite none of the foci being >1 mm in size, and biology of the microinvasive tumor cells, it is our opinion that patients with microinvasive disease in the breast associated with lymph node metastasis should be treated similarly to patients with small invasive carcinoma (pT1a), particularly those with macrometastatic nodal disease.

This study has several strengths and limitations. The strengths are mostly related to the rarity of this disease and the limited meta-analyses published in the literature. Furthermore, this is the first meta-analysis that investigates the prognostic factors of microinvasive breast carcinoma and found that lymph node status is effectively associated with prognosis in terms of DFS. However, this study has some limitations. First, the number of papers considered is very low; this is mainly due to the limited number of studies specifically focusing on the prognostic factors of microinvasive breast carcinoma. Second, we adopted strict inclusion criteria: articles were selected only if HRs were directly available in papers and/or their supplementary materials. Even if currently recognized as the best indicator of outcome in time-to-event data [45], HR is often not clearly expressed in studies and clinical trials [46]. Some researchers have developed methods to calculate HRs from other data [45]; however, we preferred to strictly limit the search to articles with clearly identifiable HRs. Third, another difficulty may be represented by the ability of pathologists to recognize microinvasive breast carcinoma (by missing microinvasion or macroinvasion), which may lead to both overdiagnosis and underdiagnosis of this entity [47]. Fourth, the very low metastatic rate of microinvasive carcinomas may have reduced the impact of our analysis; however, statistical analyses have been performed and calculated in all comparisons. Finally, the studies included in the meta-analysis for lymph node status did not define positive lymph nodes in terms of micro- vs. macrometastases. Lymph node metastases are subject to differences in interpretation and detection methods, influencing the rate and prognostic impact of nodal positivity [48,49]. The only prognostic factor identified as meaningful in microinvasive carcinomas, lymph node status, reflected by the simple variable “node-positive” vs. “node-negative”, may hinder different rates of isolated tumor cells, micrometastases, macrometastases, and occult metastases, adding some confounding lack of details to the clarified prognosticator. However, this is still the best estimation of prognosticators and the highest level of evidence that can be gained from published data. This study may also help better define future studies and examine prognostic factors more extensively, with a specific focus on microinvasive breast carcinomas. 

## 5. Conclusions

This systematic review and meta-analysis showed that lymph node status is a prognostic factor of microinvasive breast carcinoma and directly affects DFS: positive lymph node status worsens DFS. The other prognostic factors examined (ER, PR, HER2, multifocality of invasion, grade, and patient’s age) were not significantly associated with DFS. Given its important prognostic role, it may be useful to remove and examine sentinel lymph nodes in microinvasive breast carcinoma. These features may also help better define the disease and its therapeutic strategies.

## Figures and Tables

**Figure 1 cancers-15-03007-f001:**
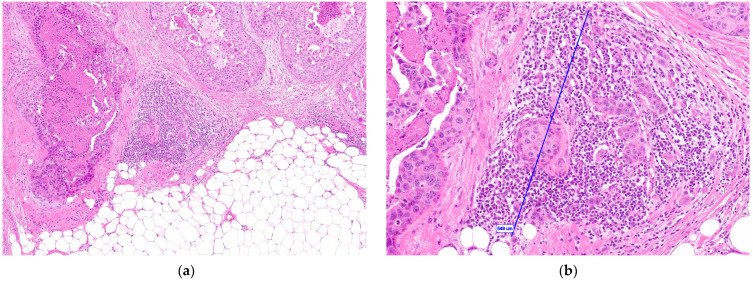
Microinvasive breast carcinoma: (**a**) at low-power view (10×), it is typically found adjacent to ductal carcinoma in situ, comedocarcinoma type; (**b**) at high-power view (40×), the invasive component measures ≤1 mm (in this example, 549 μm = 0.549 mm, as indicated by the blue line).

**Figure 2 cancers-15-03007-f002:**
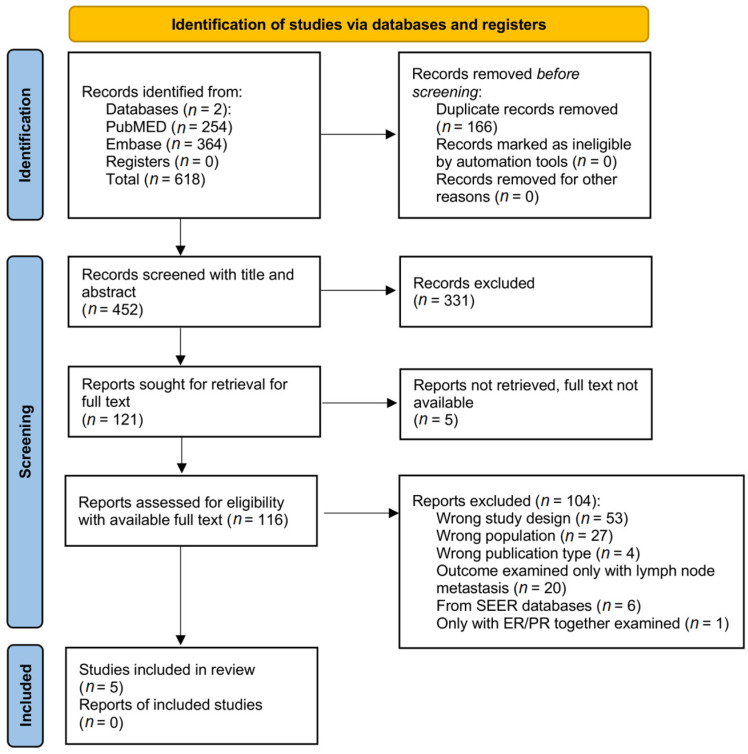
The PRISMA flow chart describes the selection of articles.

**Figure 3 cancers-15-03007-f003:**
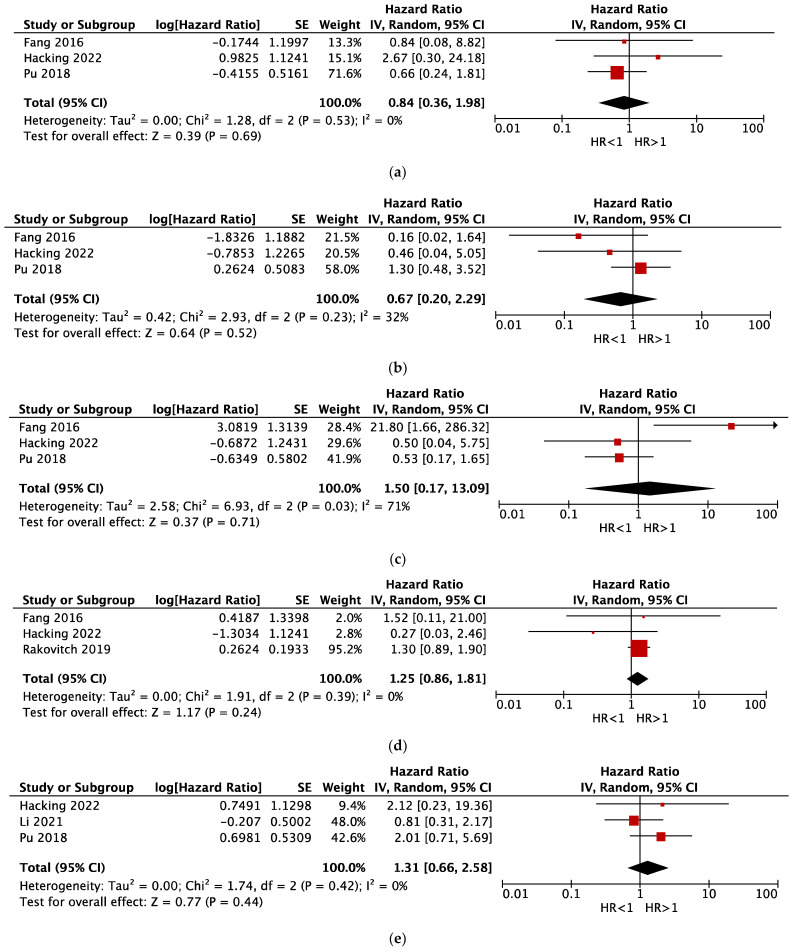
Forest plots of comparison for prognostic factors on DFS in microinvasive breast carcinoma. Red squares represent HRs of single studies; black diamonds represent the pooled results of all studies. If HR is >1, the prognostic factor is associated with a worse prognosis (lower DFS). (**a**) ER; (**b**) PR; (**c**) HER2; (**d**) multifocality of invasion; (**e**) grade; (**f**) age; and (**g**) lymph node status. Only lymph node status was suggested to be associated with prognosis (*p* = 0.05; positive lymph node status worsened prognosis and reduced DFS). Legend: DFS, disease-free survival; HR, hazard ratio; ER, estrogen receptor; PR, progesterone receptor; [12,29,30,31].

**Table 1 cancers-15-03007-t001:** Risk of bias according to the QUIPS tool. Legend: Ref., reference.

Author	Year	Ref.	1. Study Participation	2. Study Attrition	3. Prognostic Factor Measurement	4. Outcome Measurement	5. Study Confounding	6. Statistical Analysis and Reporting
Fang	2016	[29]						
Hacking	2022	[30]						
Li	2021	[12]						
Pu	2018	[31]						
Rakovitch	2019	[32]						
	Low risk							
	Moderate risk							
	High risk							

**Table 2 cancers-15-03007-t002:** Summary of the clinicopathological features of the five selected articles. Legend: Ref., reference; N., number; HR, hazard ratio; ER, estrogen receptor; PR, progesterone receptor; DCIS, ductal carcinoma in situ; DFS, disease-free survival.

Author	Year	Ref.	Country	N. of Cases	Period	Prognostic Factors Examined	HR Calculated on	Type of Analysis
Fang	2016	[29]	China	84	2002–2014	N. of foci, ER, PR, HER2	DFS	Multivariate
Hacking	2022	[30]	USA	72	2010–2020	DCIS size, N. of foci, nuclear grade, age, ER, PR, HER2, SLNB, surgery, radiation status	DFS	Univariate ^1^
Li	2021	[12]	China	1286	2008–2019	Surgery, tumor volume, grade, Ki67, age, lymph node status, margin, chemotherapy, radiotherapy	DFS	Univariate (also multivariate for some of them)
Pu	2018	[31]	China	242	1997–2014	Age, tumor size, lymph node status, grade, necrosis, ER, PR, HER2, Ki67, chemotherapy, radiotherapy, endocrine therapy, trastuzumab, therapeutic schemes	DFS	Univariate (also multivariate for some of them)
Rakovitch	2019	[32]	Canada	267	1994–2003	N. of foci of microinvasion	DFS	Multivariate

^1^ Multivariate values were also available, but their confidence intervals were so high that it was necessary to consider univariate values.

**Table 3 cancers-15-03007-t003:** Summary of meta-analyses performed. The significant values are highlighted in bold. Legend: ER, estrogen receptor; PR, progesterone receptor.

Prognostic Factor Examined	References Included	Total n. of Cases	*I* ^2^	Z	*p*	GRADE Quality of Evidence
ER	Fang 2016 [29], Hackin 2022 [30], Pu 2018 [31]	398	0%	0.39	0.69	Moderate
PR	Fang 2016 [29], Hackin 2022 [30], Pu 2018 [31]	398	32%	0.64	0.52	Moderate
HER2	Fang 2016 [29], Hackin 2022 [30], Pu 2018 [31]	398	71%	0.37	0.71	Low
Multifocality of invasion (≥2 foci)	Fang 2016 [29], Hackin 2022 [30], Rakovitch 2019 [32]	423	0%	1.17	0.24	Moderate
Grade (1/2 vs. 3)	Hackin 2022 [30], Li 2021 [12], Pu 2018 [31]	1613	0%	0.77	0.44	Moderate
Age (≥50)	Fang 2016 [29], Hackin 2022 [30], Pu 2018 [31]	398	62%	0.88	0.38	Low
Lymph node status	Li 2021 [12], Pu 2018 [31]	1528	28%	1.94	**0.05**	Moderate

## Data Availability

No new data were created. Summaries of all meta-analyses are provided in the Tables and Figures.

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
