# Peer review of "Prognostic Markers of Microinvasive Breast Carcinoma: A Systematic Review and Meta-Analysis"

_cancers, 2023, doi:10.3390/cancers15113007_

Round 1

Reviewer 1 Report

This is a very concise systematic review and meta-analysis which could be of better benefit for readers if more information is added.

Criticisms:

1)      Readers would benefit by the inclusion of essential data characterising this disease: DFS, survival rate and distant metastasis rate, to establish the need for its investigation.

2)       Authors claim that: “the prognostic factors of microinvasive breast carcinoma have not been  extensively or uniformly examined in the literature”. Does this refer to research studies or systematic reviews? Or both? These essential facts are presented very vaguely. It would be immensely helpful for readers if authors could be more precise and elaborate in more length on the previously published meta-analyses involving microinvasive carcinoma and their results.

3)      Microinvasive carcinoma is relatively rare and thus not so thoroughly studied. The other impediment to its study is its very low metastasis rate. Taken together, it is hard to obtain a reliable statistical analysis.  It would be informative for readers if authors elaborate on the above facts and discuss these issues based on the published literature. 

English writing is satisfactory

Author Response

We are grateful to the editors and reviewers, their interest in our work, and for the valuable comments that have helped to improve the quality of our paper. We have considered all the comments and suggestions and have made the changes to the manuscript, as listed in the attached file.

We have highlighted our responses in this document by using red fonts.

We have highlighted the changes in the manuscript by using the “track Changes” function on the Word file as requested.

Reviewer 2 Report

The manuscript was prepared very well. The introduction section justifies the purpose of the study. I congratulate the authors for the preparation of the manuscript

I would like to congratulate the authors for the structure of the manuscript and all the research carried out. It is highly publishable. However, there are some concerns, in part

Introduction

·       Why is this study considered relevant?

·       I suggest you incorporate a little more information related to Prognostic markers (can help doi:10.3390/diagnostics10070443)

·       Figure 1 is your own creation?

·       describe potential side effects

Methods

·       The methodology does not require any suggestions the procedure is methodologically perfect

Results

·       They are perfectly described, it is the strong part of the manuscript

Discussion

·       Include a section on strengths / limitations.

·       Is it possible to describe more mechanisms responsible for the described actions?

·       What does this article contribute to, the authors should make their own assessment and include their own discussion of the results shown in the manuscript?

·       include a section on future scenarios

Conclusion

·        In the Conclusion section, state the most important outcome of your work. Do not simply summarize the points already made in the body — instead, interpret your findings at a higher level of abstraction. Show whether, or to what extent, you have succeeded in addressing the need stated in the Introduction (or objectives).

Author Response

We are grateful to the editors and reviewers, their interest in our work, and for the valuable comments that have helped to improve the quality of our paper. We have considered all the comments and suggestions and have made the  changes to the manuscript, as listed in the atteched file.

We have highlighted our responses in this document by using red fonts.

We have highlighted the changes in the manuscript by using the “track Changes” function on the Word file as requested.
